# Iron: Not Just a Passive Bystander in AITD

**DOI:** 10.3390/nu14214682

**Published:** 2022-11-05

**Authors:** Michał Szklarz, Katarzyna Gontarz-Nowak, Wojciech Matuszewski, Elżbieta Bandurska-Stankiewicz

**Affiliations:** Clinic of Endocrinology, Diabetology and Internal Medicine, School of Medicine, Collegium Medicum, University of Warmia and Mazury in Olsztyn, 10-957 Olsztyn, Poland

**Keywords:** AITD, thyroid, iron deficiency

## Abstract

Autoimmune thyroid disease (AITD) is the most prevalent autoimmune disease all over the world and the most frequent cause of hypothyroidism in areas of iodine sufficiency. The pathogenesis of AITD is multifactorial and depends on complex interactions between genetic and environmental factors, with epigenetics being the crucial link. Iron deficiency (ID) can reduce the activities of thyroid peroxidase and 5′-deiodinase, inhibit binding of triiodothyronine to its nuclear receptor, and cause slower utilization of T3 from the serum pool. Moreover, ID can disturb the functioning of the immune system, increasing the risk of autoimmune disorders. ID can be responsible for residual symptoms that may persist in patients with AITD, even if their thyrometabolic status has been controlled. The human lifestyle in the 21st century is inevitably associated with exposure to chemical compounds, pathogens, and stress, which implies an increased risk of autoimmune disorders and thyroid dysfunction. To summarize, in our paper we discuss how iron deficiency can impair the functions of the immune system, cause epigenetic changes in human DNA, and potentiate tissue damage by chemicals acting as thyroid disruptors.

## 1. Introduction

Autoimmune thyroid disease (AITD), chronic autoimmune thyroiditis, also known as chronic lymphocytic and Hashimoto’s thyroiditis, was first described by Hashimoto in 1912. Hashimoto reported four patients with goiter, lymphocytic infiltration on histopathological examination, parenchymal cell atrophy, fibrosis, and eosinophilic inclusions in some follicular cells [1]. Until now, several variants of Hashimoto’s thyroiditis have been described, which are different from the original disease reported by Hashimoto [2]. Classically, the disease manifests with painless enlargement of the thyroid gland (goitrous form) with or without hypothyroidism in young or middle-aged women. The atrophic form is less common and is usually diagnosed with serological tests in patients with hypothyroidism and normal or atrophic thyroid. AITD is the most prevalent autoimmune disease all over the world and the most frequent cause of hypothyroidism in areas of iodine sufficiency [3]. AITD results in hypothyroidism in 20–30% of patients [4,5]. AITD affects 0.3–1.5/1000 subjects/year and is 4–10 times more frequent in women than in men [(3.5–5/1000 subjects/year in women versus 0.6–1.0/1000 subjects in men [6]. AITD is diagnosed based on increased serum levels of antithyroid antibodies and/or hypoechogenicity of the thyroid gland at ultrasound [4,5]. Studies indicate that anti-TPO and/or anti-Tg antibodies are found in 2–17% of women of childbearing age.

Iron is essential for proper functioning of all cells in the human body. It is involved in the transfer of oxygen and electrons and is crucial for the activity of numerous enzymes [7]. Two percent of human genes encode iron-binding proteins and 6.5% of all enzymes are directly dependent on iron [8]. 

Iron deficiency (ID) is among the most prevalent nutritional deficiencies, affecting as many as 2 billion people all over the world, mostly pregnant women and children [9]. Its prevalence correlates with socioeconomic status –iron deficiency is estimated to affect 4–18% of the population in the United States and Northern and Western Europe, 9–50% of the population in Eastern Europe, 64% in Asia, 54% in Southern Asia, and 62% in Latin America [10]. It is the underlying cause of anemia in 42% and 50% of anemia cases in children and women, respectively. ID is also among the five most frequent causes of disability in women from 35 countries [11].

Several chronic diseases are frequently associated with iron deficiency anemia, notably chronic kidney disease, chronic heart failure, cancer, and inflammatory bowel disease [10]. Moreover, iron deficiency anemia (IDA) is an independent risk factor for increased perioperative mortality [12]. 

ID becomes more prevalent in patients with a higher NYHA (New York Heart Association) class, increases the mortality and morbidity of patients with HfrEF (heart failure with reduced ejection fraction), and impairs exercise tolerance [13]. It can impair brain functions. Iron is essential for the synthesis of neurotransmitters [14], including for the activity of tryptophan hydroxylase (involved in serotonin production) and tyrosine hydroxylase (involved in norepinephrine and dopamine production). ID results in irreversible disorders of dopamine release in young rats [15]. Georgieff was the first to show that ID impairs hippocampal metabolism, dendritogenesis and development [16]. Other studies have reported that behavioral changes are associated with hippocampal dysfunction, including spatial learning disorders [17] and impaired trace conditioning [18]. ID can increase the risk of psychiatric disorders, including depression and psychotic disorders [19]. 

As mentioned above, iron deficiency is the most common nutritional deficiency worldwide, with particularly high prevalence in women of reproductive age. Due to high opportunities for iron deficiency prevention and easy iron supplementation, iron metabolism should be investigated in relation to AITD–as the disease can cause not only hypothyroidism but also infertility and can increase the frequency of obstetric complications in euthyroid women [20].

The aim of this paper is to discuss the effects of iron deficiency on the development of autoimmune thyroid disease based on the available evidence. 

## 2. Material and Methods

The study presents an analysis of data in the currently available literature. We examined electronic databases, including: MEDLINE and Pubmed. The search terms we used included: “iron deficiency”; “iron deficiency anemia”; “iron and thyroid disease”; “iron and autoimmune diseases”; “iron deficiency and AITD”; “iron and obesity”; “iron and diabetes”; “iron and immunology”; “iron metabolism and environmental pollution”.

## 3. Etiology of AITD

### 3.1. AITD Is Associated with Other Autoimmune Disorders

The pathogenesis of AITD is multifactorial and depends on complex interactions between genetic and environmental factors leading to the breakdown of immune tolerance [21]. Genetic factors are estimated to contribute to the disease in 70%, while environmental factors, associated with the activation of the innate immune system, in 20–30% [22]. In numerous cases AITD appears to be the first manifestation of the antiphospholipid syndrome (APS) [23]. Studies confirm that AITD patients are more likely to suffer from Huntington’s disease, multiple sclerosis, celiac disease, diabetes, sickle-cell anemia, sarcoidosis, alopecia areata, rheumatoid arthritis, polymyalgia, articular psoriasis, scleroderma, and HCV cryoglobulinemia [24]. 

### 3.2. Impact of Genetic Factors on the Development of AITD

Genes responsible for the development of AITD are genes that encode immune regulatory factors and are involved in the complex processes of ensuring robust immune responses against appropriate foreign antigens while maintaining tolerance to self-antigens [21]. AITD may develop as a result of congenital alterations in the development of central tolerance in the thymus and Treg-dependent peripheral tolerance, which can lead to abnormal co-stimulation of T cells and precursor cells (PCs) in the immunological synapse [21].

AITD develops in 20–30% of siblings of patients with AITD, with the sibling risk ratio of 16.9. Concordance rate for AITD in monozygotic twins is estimated to be 29–55% [25]. Simmonds et al. found that AITD susceptibility is associated with genes: IL2RA, HLA, PTPN22, and CTLA4 [26,27]. Other genes that increase the risk of AITD include: FCR3, TSH-R, HLA-1, FOXE3, GDCG4p14, and RNASET2, expressed on CD4+ and CD8+ cells; 7 out of 11 AITD susceptibility genes are involved in maintaining proper functions of T cells, which speaks for their key role in the development of AITD [5]. The first reported monogenic predisposition to AITD is splice site variant in the thyroglobulin gene (TG c. 1076-1G > C) [20,28]. 

### 3.3. Impact of Viral Infections, including COVID-19, on the Development of AITD

Viruses can activate innate immunity and lead to the development of AITD. An association was confirmed between HTLV1, HSV, Rubella, Mumps, EBV, HIV, HCV, Parvovirus, and Enterovirus infections and AITD [29,30]. ID can impair and distort the body’s immune response to viral infections. Enhanced lymphocyte metabolism requires higher amounts of iron, and mice with ID presented attenuated T- and B-cell response to Adenovirus and Vaccinia virus [31].

One of the hypotheses that explain the increased incidence of autoimmune disorders in the 21st century says that excessive hygiene lowers the threshold for immune activation [32] in response to pathogens. Numerous autoimmune diseases are triggered by infection. Reducing inflammation in autoimmune diseases can restore the immune balance and induce remission. Infections may result in disruption of the immune regulation, which may cause autoantigen mimicry. Infection can also affect target organs and increase their susceptibility to autoimmune disorders. The dual role of infection means that it is both an indicator of autoimmunity and a promoter of disease progression and escalation of the autoimmune process to the autoimmune disease. Molecular mimicry, antigen expression, and tissue modifications can lead to antigen-specific signals that initiate autoimmune reactions. Other effects include induction of the pathogen-associated molecular pattern (PAMP) and Toll-like receptors involved in the progression from the autoimmune reaction to the autoimmune disease. Infection can shift the body’s normal immune response toward a pathological process that results in full-blown disease. Effective apoptosis is a mechanism that enables the body to eliminate damaged cells without inducing inflammatory state. Infection and other environmental factors can impair apoptosis and increase the risk of autoimmune process. Damaged cells that are not cleared by macrophages and monocytes remain in tissues as a potential source of autoimmune process. Impaired apoptosis can lead to autoimmune lymphoproliferative disorders associated with CD95 (fas)-fas-ligand mutation. Iron deficiency can impair ferroptosis–a type of iron-dependent cell death.

There is a connection between the effects of COVID-19 infection and thyroiditis, which involves a mechanism of direct effects of the virus on thyrocytes, with initiation of inflammation in the thyroid gland, and possible indirect effect on the hypothalamic–pituitary–thyroid axis [33]. About 80% of patients who died of COVID-19 had functional ID due to iron sequestration as well as increased IL-6 and hepcidin levels [34]. Studies have confirmed that functional ID was associated with longer duration of hospital stay, fivefold increase in the risk of admission to intensive care unit (ICU), and eightfold increase in the risk of mechanical ventilation [34]. COVID-19 virus replication relies on the host intracellular proteins dependent on iron availability [35], including RNA reductase. Therefore, immunity to COVID-19 may directly depend on intracellular iron stores.

### 3.4. How the Inflamamtory Infiltration Develops in AITD

AITD is characterized by lymphocytic infiltration of the thyroid gland, caused mostly by T lymphocytes, which leads to the thyroid fibrosis and atrophy [36]. The CD4/CD8 cell ratio is changing due to the decrease in CD8 + count. Active CD8 and/or CD4 infiltration is often observed in the inflammatory infiltration, with evident predominance of CD4 [37]. Active T lymphocytes themselves show higher levels of HLA-DR expression. Moreover, a decrease in Treg [38] is seen, with aberration of Helios and PD-1 expression [38,39] and activation of the PD-1/PDL-1 signaling pathway. An important role is played by Th17, Th22 and recently discovered thyroid specific T follicular helper cells (Tfh) [40]. Th17 cells are abundant in the thyroid gland, producing Il-17 [40,41]. A study involving activation of dendritic cells (DCs) by exosomes has shown that exosomes of AITD patients were activated mostly by CD4+ monocytes and C11+ dendritic cells. They were different from the control group–they could present antigens for DCs and bind Toll-like receptors, causing the activation of DCs via the Nf-kappaB signaling pathway, which might be the cause of altered differentiation of CD4+ and subsequent thyroid inflammation [42]. 

The thyroid autoimmune process is strongly affected by cytokines. Patients with a high degree of lymphocytic infiltration and hypoechoic pattern of the thyroid are observed to have high levels of IFN-gamma [43]. INF gamma and TNF alpha, produced by Th1 cells, stimulate thyrocytes to release CXCL10 [43,44]. CXCL10, binding to chemokine receptor (C-X-X- motif receptor CXCR3) on Th1 cells, makes Th1 cells migrate to target tissues and promote inflammation. Tfh cells with CXCR5 expression, which produce IL-21, are necessary for activation of B cells and formation of germinal centers in the thyroid [44]. It is intrathyroidal B lymphocytes that are responsible for the production of antibodies. In vivo studies have shown that antithyroid antibodies come from the thyroid, juxta thyroid lymph nodes, and other germinal centers in the bone marrow. 

### 3.5. ID and Production of Anti-TPO Antibodies

Iron deficiency can trigger inflammatory processes in the thyroid gland, including the development of thyroid antibodies. The pathogenesis of AITD involves interactions between the patients’ serum and a number of antigens. Thyroglobulin (Tg) gene product represents 80% of the total thyroidal protein. Having been synthetized, Tg can pass to the systemic circulation, where it is exposed to the immune system [3]. Thyroid peroxidase (TPO) antibodies can recognize predominantly conformational epitopes in the immunodominant region of the antigen, which contains two overlapping regions, A and B, that become the major target for anti-TPO antibodies [45].

Anti-TPO antibodies stimulate two types of cytotoxicity: antibody-dependent cytotoxicity and complement-dependent cytotoxicity, which results in thyroid atrophy [46]. The relationship between IDA and autoimmune disorders is more pronounced in women. Its risk factors include arterial hypertension, dyslipidemia, cancer, allergic rhinitis, chronic obstructive pulmonary disease (COPD), urticarial, and chronic liver disease. Another important risk factor is patient’s age, with the highest risk observed in subjects aged 20–40. In the group of patients aged 20–40 years, approximately 65% are likely to develop an autoimmune disease within 5 years of being diagnosed with IDA [47]. In a study of 2581 pregnant women anti-TPO antibody level was higher in subjects with iron deficiency, with no differences in TT4 levels between study groups. Hypothyroidism and subclinical hypothyroidism were more prevalent in IDA group versus mild or no iron deficiency groups. The study has shown that the incidence of AITD was growing along with the severity of iron deficiency. In a meta-analysis by Luo et al. from 2021, ID in women of reproductive age resulted in a twofold increase in the risk of elevated titers of TPO- and/or Tg-antibodies [48].

### 3.6. Iron Function in the Synthesis and Tissue Actions of Thyroid Hormones

Iron is crucial for proper TPO activity, which is the key enzyme involved in biosynthesis of thyroid hormones (Figure 1a,b). Iron deficiency can reduce the activity of thyroid peroxidase (TPO). Studies have shown that the administration of hemin can increase the level and activity of TPO from 20% to 120% [49], while the administration of succinylacetone can decrease its activity by 25–37% [49]. A study with seven groups of rats showed that iron deficiency can reduce the activity of TPO. Three groups (ID-3, ID-7, ID-11) were administered food low in iron (3, 7 and 11 ppm iron, respectively). The other four groups were provided adequate iron intake (35 ppm). After 4 weeks, Hb, T3, and T4 levels were considerably lower in groups with ID, and TPO activity was reduced (by 56%, 45% and 33%, respectively) [50]. A study with human subjects confirmed a reduced activity of TPO in groups with ID (by 33–56% depending on the severity of iron deficiency) [51]. ID resulted in reduced activity of 5′-deiodinase [52] and slower utilization of T3 from the serum pool [53]. T3 binding to its nuclear receptor is also decreased [54] and iodine prophylaxis is less effective in ID. In Figure 2 we have presented the summary of the impact of iron deficiency on thyrometabolism (Figure 2). 

## 4. Iron Deficiency Impairs the Body’s Immune Response and Alters the Human Microbiome

Iron is necessary for immune response reactions. TRF1 (a protein responsible for iron transport into the lymphocytes) gene mutation can lead to immune deficiency states in humans, with low IgG levels and reduced B-cell and T-cell proliferation [55]. Iron is of key importance for appropriate activities of peroxidases and synthases involved in nitric oxide generation, which is crucial for maintaining proper functions of immune cells. Moreover, iron is involved in the regulation of cytokine production and signaling pathways [56]. ID can affect cytokine expression on lymphocytes, leading to an increase in cells with IFN-gamma expression and a decrease in cells with IL-4 expression [57]. Recent studies have shown the role of cytokines and chekokines in the pathogenesis of AITD. In thyroid tissue Th1 may be responsible for enhanced INF gamma level which stimulates CXCL10 production from the thyroid. Patients with a high degree of lymphocytic infiltration and a hypoechoic thyroid pattern are observed to have high levels of IFN-gamma [43]. INF gamma and TNF alpha, produced by Th1 cells, stimulate thyrocytes to release CXCL10 [43,44]. Iron deficiency impairs the TLR4 pathway signaling [58] and affects the activity of iron-dependent ubiquitin ligase [59] and regulation of SHP1 tyrosine phosphatase [60]. Nevertheless, iron deficiency can lead to the activation of the signaling pathways dependent on NF-kappaB, transcription factors necessary for the expression of genes crucial for innate and adaptive immunity, which may exacerbate inflammatory processes in the thyroid [61]. Dendritic cell activation involves the Nf-kappaB signaling pathway, which might be the cause of altered differentiation of CD4+ and subsequent thyroid inflammation [42]. 

Adequate cell-mediated immune response is delayed in ID [62]. A number of studies in humans have shown an impairment of innate immunity in subjects with ID. Bactericidal activity of macrophages is reduced, while neutrophils contain lower amounts of myeloperoxidase (MPO), involved in the generation of reactive oxygen species and responsible for intracellular pathogen death [63]. Children with ID had lower levels of IgG and IL-6, and decreased phagocytic activity with reduced oxidative burst in neutrophils [64]. ID affects blastogenesis and mitogenesis of T cells and protein kinase C activity [65]. In mice it can lead to a decrease in T-cell mediated antigen specific response, production of antibodies, and B-cell proliferation [66]. Moreover, a reduction in the induction of cyclin S, and delayed entry of B cells in phase S of the cell cycle during B-cell proliferation is observed [66]. Histone demethylation is of key importance for T-cell differentiation. Demethylation of cyclin E1 of histone H3K9 is iron dependent, which in case of iron deficiency may impair T-lymphocyte differentiation [67]. The most severely affected in ID was TLR-dependent B-lymphocyte proliferation, followed by B-lymphocyte response to BCR stimulation and relatively least affected T-lymphocyte response to TCR stimulation [66]. An effective immune response including self-tolerance, cell-mediated immunity, and humoral immunity plays a critical role in the development of AITD. The dysfunctional adaptive immunity response in iron deficiency states may favor the production of antithyroid antibodies by intrathyroidal B lymphocytes. Moreover, the Th1- and Tfh-prevalent autoimmune response is characteristic for AITD, which is also affected to a considerable extent by ID in the mechanisms of impaired response to the TCR transduced stimulation and disturbed T-lymphocyte differentiation.

AITD can lead to small intestinal bacterial overgrowth (SIBO) with altered composition of intestinal microbiota [68]. A hypothesis suggests that microbiota dysbiosis can lead to AITD. Microbiome can affect iron absorption, which in turn can alter composition and metabolic activity of microbiome and modulate intestinal effects on the immune system [69]. A study has shown that, in the absence of viable intestinal microbiota, iron absorption decreased by as much as 25% [70]. Moreover, ID can result in higher sensitivity of body cells to bacterial endotoxins [71]. Of course, we must keep in mind that patients with AITD are more likely to suffer from celiac disease and gastritis—for this reason we cannot exclude that the higher prevalence of iron deficiency in AITD patients partially results from malabsorption disorders [72]. Moreover, celiac disease is finally confirmed in 1 out of 31 patients undergoing diagnostic procedures because of iron deficiency. Therefore, in case of concomitant ID and AITD, we recommend diagnostic procedures for celiac disease and gastritis. 

To summarize, the effect of iron deficiency on the body’s immune reactions, including the breakdown of immune tolerance, dysfunction of T lymphocytes, and distorted humoral response, can underly the possible mechanisms leading to the development of AITD. 

## 5. Iron Deficiency and Thyroid Sensitivity to Environmental Factors

Iron homeostasis disruption can lead to higher sensitivity of the body to environmental pollutants. Functional groups of chemical compounds remove iron atoms from inside the cell, leading to its functional deficiency. Functional ID activates kinases and transcription factors able to trigger inflammatory reactions. Increased iron availability can diminish its cellular deficiency and counteract biological effects of chemical particles [73]. Iron is kinetically preferred by chemical particles due to its electropositivity, high affinity for oxygen-containing groups, and availability. Due to endocytosis of inorganic particles, their surface functional groups come into reaction with iron to produce chemical complexes [74]. Chemical particles have the continuous ability to export iron from cells, while the body responds to it with iron sequestration to such an extent to make it unavailable to chemicals. Therefore, toxic ferruginous bodies are formed [75].

Polychlorinated biphenyls (PCBs) including bisphenol A, phthalates, brominated flame retardants, and perfluorinated compounds can disrupt thyroid function [76]. Thyroid function can also be disturbed by cadmium, which accumulates in thyrocyte mitochondria [77], or by manganese, which may suppress TSH release by inhibiting dopamine secretion [78]. A study of 236 women, conducted by Benven S et al. [79] has shown higher titers of anti-TPO and anti-Tg antibodies in women from Group A who ingested swordfish versus Group B who received omega 3 dietary supplements. In Group A, mercury intake was 1000 mcg/month versus 25 mcg/month in Group B. An in vitro study by Fallahi et al. [80] investigated the effect of vanadium oxygen on thyroid growth and proliferation and secretion of CXCL8 and CXCL11 in thyrocytes. Animal studies have shown that iron deficiency may increase tissue sensitivity to the toxic effects of heavy metals including cadmium and manganese [81]. Vanadium oxygen (V205) was able to induce secretion of CXCL8 and CXCL11 by thyrocytes. This can trigger and maintain inflammatory processes in thyroid. In a study by Sun X et al. which involved 675 pregnant women, an exposure to heavy metals (vanadium, arsenic) was associated with lower levels of thyroid hormones and decreased birth weight of children [82]. Populations exposed to vanadium were observed to have higher incidence of thyroid cancer, for example in the Etna region. Vanadium oxygen can increase chemokine secretion by Th1 cells, synergically potentiating the effects of cytokines such as INF gamma and TNF alpha, which results in induction of inflammatory state in the thyroid gland [80]. Exposure to vanadyl sulphate results in a reduction of iron in the cellular nucleus and mitochondria. In response to the vanadyl sulphate induced iron loss, the iron import into the cell is increased in order to compensate for these sudden deficiencies. Intracellular iron deficiency leads to increased DMT1 expression. Vanadium has the ability to bind with iron in transferrin, lactoferrin and ferritin. Following vanadium exposure, the amount of non-heme iron in the nucleus and mitochondria decreased rapidly with its concomitant increase in the cellular supernatant [83]. Vanadium is able to interact with iron incorporated in iron body stores [84,85,86,87]. Similar to the biological effects of vanadium exposure, iron dependent mitochondrial proteins can be targeted by other heavy metals, which may exacerbate their toxicity and this effect is potentiated by iron deficiency.

A twofold increase in the incidence of thyroid lesions suspected for malignancy was observed in FNAB in polluted areas. Arena S et al. [88] studied a group of 391 patients undergoing thyroid FNAB in areas of petrochemical industry (Group A) and compared them to a control group of 622 patients living at a distance from exposure areas (Group B). Lymphocytic thyroiditis and suspected malignant lesions were observed considerably more frequently in Group A. In a Brazilian study by de Freitas CU et al. [89], higher anti-TPO titers were seen in patients exposed to petrochemical industry. Freire C et al. [90] in their study of inhabitants of industrial areas polluted with organochlorine (OC) pesticides found higher anti-TPO titers following methoxychlor exposure. Cells administered iron prior to their exposure to pollutants demonstrated lower oxidant levels. Exposure to silica resulted in increased levels of IL-6 and IL-8, although these cytokines were not released by cells rich in iron [86]. Studies have shown that silica exposure increased NF-kappaB activation at least fivefold. Mitochondrial sources of iron were complexed by silica which was then centrifuged into the nuclear fraction. This illustrates how chemical particles can disrupt the intracellular iron homeostasis.

Tissue damage, including the damage to the thyroid gland, is the final outcome of iron deficiency during their exposure to chemical compounds. Iron deficiency states result in an increased activity of MAP-kinases and transcription factors, which potentiates inflammatory processes in tissues. In response to the exposure to chemical particles iron import into cells is upregulated. Tissue damage is particularly likely to occur in cases of exposure to nanomolecules because of their small sizes and high interaction surfaces, which potentiates biological effects of iron deficiency. Nanomolecules may contain alcohol, diol, epoxide, ether, dibenzofurans, and benzodioxines [73]. In a Slovakian study, increased thyroid volume and higher incidence of thyroid dysfunction were seen in employees exposed to PCBs and polychlorinated dibenzodioxins and dibenzofurans [91], which confirmed positive correlation between AITD and the severity of OC exposure. Similarly, employees exposed to polybrominated diphenyl ethers and polybrominated biphenyls (PBBs) had elevated titers of thyroid antibodies and higher TSH levels [92]. Schell LM et al. [93] conducted a study of 115 adolescents from the Akwesasne Mohawk Nation living on the territory near the St. Lawrence River in the State of New York. For many years, waters of the St. Lawrence River and its three tributaries had been polluted by aluminum foundries. As a result, the PCBs dichlorodiphenyldichloroethylene (p,p’-DDE), hexachlorobenzene (HCB), and Mirex (an organochlorine insecticide) were included into the local food chain. Adolescents who had been breastfed in their infancy were found to have elevated anti-TPO levels. Vietnam veterans exposed to Agent Orange (2,4—dichlorophenoxyacetic acid and tetrachlorodibenzodioxin (TCDD)) were more likely to suffer from diabetes mellitus and a number of thyroid and pituitary disorders, with threefold increase in the incidence of Graves’ disease [94]. AITD was considerably more prevalent in regions around Chernobyl, affecting 6.4% of children and adolescents versus 2.4% of those living in regions not exposed to radioactive fallout [95]. In a study by Tajtakova M et al. [96] in which 324 children from regions of high exposure to nitrogen were compared to 596 children from control groups, anti-TPO and TSH levels were higher and thyroid hypoechogenicity was more frequent in nitrogen-exposed subjects. A study of 70 vitiligo patients conducted by Colucci R et al. [97] has shown that exposure to PCBS was associated with increased levels of IgG antibodies to T4, while exposure to nitrates, nitriles, and soy isoflavones was associated with increased levels of IgM and IgG antibodies to T3. Studies in mice have shown that animals with ID exposed to cigarette smoke had higher levels of alveolar macrophages, and were subject to more rapid changes in the lungs [98,99]. Animals with ID presented more severe intracellular inflammation in response to smog exposure [73]. 

## 6. Impact of Iron Deficiency on Residual Symptoms in Euthyroid Patients with Aitd 

About 10–15% of hypothyroid patients complain of residual symptoms such as fatigue, lower quality of life, impaired cognitive functions and memory problems in spite of their thyrometabolic status being well-controlled [100]. In their study of 5000 subjects Ettleson et al. found that 79% of hypothyroid patients may present symptoms of “brain fog” [101]. Hypotheses that explain the development of „brain fog” in hypothyroidism focus on autoimmune processes, oxidative stress, and alterations in neurotransmitter levels [102]. Wet et al., in the their study of 199 subjects, confirmed a correlation between the presence of anti-TPO antibodies and an increased risk of depression and anxiety [103]. A study of AITD euthyroid mice has shown memory deficits, more prominent loss of synapses and astrocytes in the hippocampus and considerable reduction of neuroplasticity [104]. In a controversial study by Guldvog et al., a marked improvement in quality of life was achieved in patients with AITD following thyroidectomy [105]. Due to these residual symptoms, numerous patients with AITD seek alternative treatments, such as the use of triiodothyronine alone or desiccated porcine thyroid extract (DTE). As already mentioned, ID can result in impaired synthesis of neurotransmitters in the brain and reduced neuroplasticity. In a Japanese study of 11,876 participants, depression and experience of stress were more prevalent in the group of IDA subjects [106]. A Finnish study showed that iron can influence effectiveness of the hypothyroidism treatment. An experiment included 25 female patients with persistent symptoms of hypothyroidism in spite of being euthyroid. After 6–12 months of treatment with oral iron products the symptoms were relieved. At the onset of the treatment none of the women suffered from anemia, but all of them had ferritin levels < 60 mcg/L. Achievement of ferritin levels > 100 mcg/L resulted in reduction of residual symptoms in two-thirds of the studied patients [107]. Findings of these studies suggest that the role or iron is underestimated in AITD patients in everyday practice. As shown in Figure 2, ID can impair the body’s resistance to stress, increase sympathetic stimulation by reducing the utilization of norepinephrine and decrease tissue oxygenation. The beneficial effects of treatment with iron products may also result from the improvement of iodine utilization by the thyroid gland, and from better utilization and availability of thyroid hormones at the tissue level (including in the brain) due to the increased deiodinase and TPO activity and increased T3 binding to its nuclear receptor as well as due to the immunomodulatory effects of iron. 

## 7. Epigenetics in AITD Development

Epigenetics can be a link that integrates the effects of environmental and genetic factors on the development of AITD [108]. The most recent research shows that environmental factors can influence genes by epigenetic modifications [109]. The most common type of genetic modifications is DNA methylation. Global hypomethylation of DNA in AITD can be responsible for overexpression of some genes associated with immune system modification, and for immune cell activation that results in autoimmune attack on the thyroid tissue [110,111]. Polymorphisms in DNA methylation-regulating genes (DNMT1, MTRR) were correlated with DNA hypomethylation observed in AITD [112]. Histones play an important role in the modulation of immune response and tolerance [113]. Fragments of DNA released from injured thyroid cells can be recognized by histone H2B, causing an activation of immune response genes with subsequent potentiation of autoimmunity to thyrocytes [48]. IFN-alpha could induce changes in Tg gene expression and trigger AITD by enhancement of Lys-4 residue methylation in histone E3 at the promoter area of Tg gene [114]. IFN-alpha plays an important role in immune response to viral infection as it stimulates an increase in H3K4me3 and H3K4me1 levels in thyrocytes [115]. Saramaru et al. reported that rs3758391 and rs746720 alleles in the gene of sirtuin 1—an enzyme involved in histone deacetylation—were associated with higher titers of autoantibodies in AITD [116].

miRNA is a form of small, non-coding RNA, composed of 18–25 nucleotides, with the ability to regulate gene expression, controlling up to 60% of RNA [117]. RNA miR223-3p and miR-155-5p regulate functions of the immune system [118,119]. Mi-R-155-5p and miR-146a-5p play a particular role in immune response modulation [120,121]. MiR-146a-5p potentiates expression of interleukin-1 receptor associated kinase 1 and TNF receptor, while its reduced expression potentiates dendritic cell activation and antigen presentation [121,122]. Altered expression of miR 155-5p and miRNA-146a-5p can promote development of autoimmune diseases via breakdown of immune tolerance. Bernecker et al. showed alterations in expressions of miRNA-146a-5p and miRNA-155-5p in thyroid cells in AITD patients [123]. Zhu et al. indicated higher expression of miRNA-142-5p in AITD and its positive correlation with anti-TPO titers [124]. Overexpression of miR-142 in thyroid cells resulted in reduced claudin 1 expression and increased permeability of the thyrocyte monolayer [124]. miR-125a-3p could target the IL-23 receptor, and decreased expression of miR-125a-3p could upregulate IL-23 receptor expression in AITD [125]. Long non-coding (Lnc) RNAs composed of more than 200 nucleotides are also involved in the development of AITD [126]. Lnc RNAs can regulate gene expression by epigenetic regulations, gene transcription, post-translational regulation, and alterations of miRNA [127,128]. They contribute to the regulation of production and differentiation of T cells, and every type of T cell contains different IncRNA [129,130]. Lnc RNA-IFNG-AS1 was upregulated in AITD, and it was associated with the frequency of circulating Th1 cells and INF-gamma expression [131]. lnc RNA-IFNG-AS1 can regulate INF-gamma expression in human CD4+ cells and promote Th1 lymphocyte-dependent responses [131]. In women, one X chromosome is randomly inactivated, which is called XCI (X chromosome inactivation) and can result in a mosaic pattern of cells expressing genes from either chromosome [132]. XCI may be influenced by histone modifications and DNA and microRNA methylation [133]. Skewed XCI occurs when the inactivation of one X chromosome is silenced more than the other one [133,134], and this may cause disruption in gene expression and lead to the development of autoimmune disease [135]. Brix et al. noticed an increased frequency of skewed CXI in female twins with AITD [136,137] 

Iron is essential for DNA synthesis. Its deficiency may result in impaired DNA synthesis and subsequent alterations in programmed cell death [138]. ID potentiates DNA damage and may cause genomic instability, mimic radiation in damaging DNA by causing single- and double-strand breaks, oxidative lesions, or both [139]. In ID states, oxidative stress is more pronounced. An increase in oxidant levels and/or a decrease in antioxidant enzyme capacities is seen in IDA [140]. Iron deficiency can induce nuclear DNA base damage. An animal study has shown that iron deficiency increased oxidant levels in PMNs and potentiated liver mtDNA damage [141]. The activity of ribonucleotide reductase is decreased in iron-deficient cell cultures [142]; this could lead to decreased availability of deoxyribonucleotides for DNA repair. It is one of the major causes of imbalance between antioxidant enzymes and DNA breaking and repairing enzymes [143]. DNA polymerases, ribonucleotide reductases (RR), DNA glycolsylases, DNA primases, DNA helicasess, and DNA endonucleases are all enzymes that contain Fe-S clusters, which are involved in DNA replication, synthesis, and repair [142]. In ID states, increased chromosome fragility and reduced sister chromatid exchange are observed [144]. ID can disturb biogenesis and normal expression of microRNA [143]. MicroRNAs are involved in iron homeostasis through the post-translational regulation of genes related to the uptake, utilization and storage of iron. [145]. What is important, iron is crucial for the activity of co-factor of Drosha protein (Class 2 ribonuclease III enzyme) and Poly (rC)-binding protein 2 (Pcbp2). Drosha is responsible for the processing of pri-microRNAs into pre-microRNAs within the nucleus [146]. Therefore, abnormal heme biosynthesis and degradation may alter microRNA mediated DNA regulation. Furthermore, microRNA synthesis and expression depend on ROS and hypoxia, which are common in ID [147,148]. Hypoxia results in overexpression of Mir-373 and MiR210, with subsequent downregulation of genes involved in DNA repair: RAD 23B and RAD 52 [149]. DNA demethylation relies on the activity of iron-dependent methylcytosine dioxygenases (TET protein family), which need iron for methylcytosine-to-hydroxymethylcytosine conversion [150]. Iron deficiency can also impair histone modifications—removal of methyl groups from lysine residues is catalyzed by Jmj/AT-rich interactive domain (ARID)-containing histone demethylase (JARID) proteins, and iron is essential for their enzymatic activity [151]. 

Through the mechanisms of DNA repair and the effects of iron on the complex epigenetic interplay, iron deficiency can induce changes in the genome that potentially may promote the development of AITD.

## 8. Conclusions

The human lifestyle in the 21st century is inevitably associated with exposure to chemical compounds, pathogens, and stress, which implies an increased risk of autoimmune disorders and thyroid dysfunction. At least 30–50% of females with hypothyroidism and persistent residual symptoms in spite of adequate dose of levothyroxine can have underlying iron deficiency. Patients with AITD should undergo routine screening for ID and gastropathy. If ferritin level is lower than 70 mcg/L, diagnostic procedures for celiac disease or gastritis should be offered and iron supplementation should be started in order to restore adequate iron stores in the body and prevent tissue iron deficiency. Iron deficiency can lead to immune system dysfunction, cause epigenetic changes in human DNA, and potentiate tissue damage by chemical compounds acting as thyroid disruptors. The influence of iron deficiency on the development of AITD has been investigated in very few studies, and we believe that special attention should be paid to the issue of how ID can impair immune tolerance mechanisms. In conclusion, iron appears to be an active by-stander in the pathomechanism of AITD. 

## Figures and Tables

**Figure 1 nutrients-14-04682-f001:**
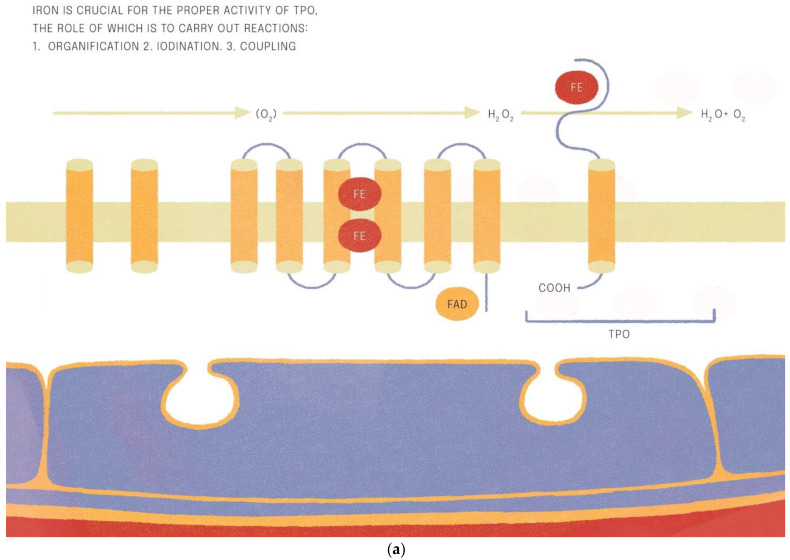
(**a**,**b**). Iron is crucial for the proper activitity of TPO (thyroid peroxidase).

**Figure 2 nutrients-14-04682-f002:**
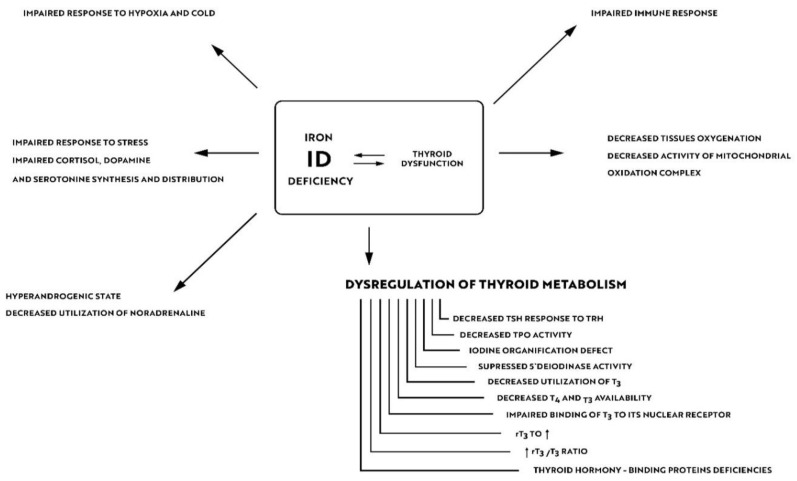
Impact of iron deficiency on thyroid metabolism.

## Data Availability

Study did not report any data.

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
