# Peer review of "Iron: Not Just a Passive Bystander in AITD"

_nutrients, 2022, doi:10.3390/nu14214682_

Round 1
Reviewer 1 Report
Authors hereby report a detailed review about the role of iron in autoimmune thyroid disease. The pathogenesis of AITD has been studied widely, but to date there are not clear data about the etiology, which is thought to be multifactorial.
Many Authors report higher risk of AITD in relation to nutrients deficiency such as Vitamin D or B12, as well as iron. The important role of these nutrients is undoubt, however in my opinion the deficiency may be the effect of hypothyroidism rather than have causative role. The modified gastro-intestinal motility and absorption which are present in hypothyroidism may play a crucial role in nutrients deficiency, even when thyroid hormone profile turn to normality once the substitutive treatment is started. It is also known that IATD may be associated with other autoimmune diseases such as gastritis or celiac disease that can impair nutrients absorption and bias more and more their speculative causative role.
Author Response
Response has been sent in the attachment.

Reviewer 2 Report
This manuscript attempts to summarize the role that iron plays in the development of AITD. However, this manuscript is so poorly organized that this relationship is barely apparent - instead, a great deal of tangential information on thyroid and autoimmune disorders is presented. This manuscript is not recommended for publication in its present state. Overall changes to organization is needed and the focus of the central message must be developed before this is position is reconsidered.
Some general comments follow, but there are much more specific comments about the studies being cited that have been withheld. These can be discussed after revisions are made.
1. The Abstract section does not adequately reflect the manuscript. Authors should increase the abstract length and incorporate some more descriptive language about the research question being addressed in the article.
2. Line 16-17: This line is not specific enough to the activities of iron, please elaborate. There are many molecules that can accomplish a "transfer of oxygen and electrons" that are also found in enzymes.
3. Line 19-23: Although iron deficiency is a problem, the authors fail to illustrate why these statistics are relevant - especially as the next sentence relates to iodine deficiency.
4. The authors fail to provide a definition of "Autoimmune thyroid disease" in the introduction. Similarly, there is no indication why iron should be investigated in relationship to AITD.
6. Line 57-95: This paragraph is too long and should be broken up into smaller concepts. In particular, lines 57-65 seem to have no direct relationship with the remainder of the paragraph and should be separated.
7. The paragraph on Lines 111-125 introduces TPO after it has already been discussed in the previous paragraphs. The information about TPO and the connection to iron should be consolidated before a discussion about a prolonged discussion about anti-TPO antibodies.
8. Figure 1 is too small and difficult to read.
9. It is unclear what section 3 brings to the greater discussion of iron and AITD.
10. The ties with AITD and iron and environmental factors (section 4) seems like a premature discussion given that the data relating iron deficiency and AITD (section 5) has not been fully discussed until later in the manuscript.
11. The first and third paragraph of section 4 does not mention iron - which argues against it being present in the article. If anything, these paragraphs should be massively truncated as it is excessively long.
12. In section 6, readers do not need a rudimentary explanation of epigenetics - however an overview of epigenetic changes that have been observed in relation to AITD is warranted.
Author Response

(The authors gave the same response as above.)

Round 2
Reviewer 2 Report
Although the revisions to the manuscript are a noteworthy improvement to the article, issues remain to make this manuscript unsuitable for publication in its present form. Some of the issues that remain are listed below. Overall, the authors need to decide if this is an article primarily about iron (with AITD subtext) or primarily about AITD (with an iron subtext) to focus the writing in the sections of the manuscript.
Major:
1. If Iron is the primary focus of the article, then keeping the order of paragraph introduction as written makes sense. Otherwise, it would follow that description(s) of AITD would be the first section of the Introduction. This would naturally lead to section 3, describing AITD before talking about iron and iron deficiency.
2. The second paragraph in the Introduction, while interesting, does not seem relevant to the rest of the information provided in the manuscript.
3. Sections 3.3 and 3.4 do not flow naturally into each other - one is speaking about the pathogenesis of AITD, while the next speaks about iron in the normal production of thyroid hormones (without mention of AITD). It is suggested that the information from 3.4 and 3.5 be re-ordered to bridge AITD into the topic of iron.
4. The first two paragraphs of section 4.1 do not seem to regard AITD at all. While this is interesting background information, it needs to be clear why this is important in AITD more explicitly - unless this manuscript is primarily about iron and its influence on the immune system rather than AITD as a central focus.
5. The interaction between viral infection, iron, autoimmune disease, and the development of AITD is poorly linked in section 4.2; iron is only mentioned in limited places - and the description of viral infections on autoimmune reactions is not targeted to topics relevant to AITD. It is also unclear if this belongs under the heading for section 4 as a whole or should be tied to section 3.
6. Section 5 is similarly poorly organized. The first paragraph speaks of iron and toxic exposures, but the second paragraph speaks of thyroid function without mention of iron. The third paragraph does mention iron, but the connections to the rest of the section are weak. The fourth paragraph integrates the topics of AITD and iron a little better, but the paragraph is long and hard to read.
7. Section 6 is also long and poorly organized. Again, the connections between iron and AITD are unclear as most of this section does not explore those interactions and instead describes T3 and T4 therapies that seem to have limited connection to iron.
8. Additionally, the role of iron in section 7 is poorly connected to epigenetics and AITD.
Minor:
1. Acronyms used should be defined more clearly. ID should be defined as Iron Deficiency on line 52. It is unclear what IDA is on line 63, NYHA is on line 65, and HFrEF on line 66.
2. TPO is first used on line 175, but defined on line 177.
Author Response
We have sent responses in the attachment.
